# Carnot Cycles in a Harmonically Confined Ultracold Gas across Bose–Einstein Condensation

**DOI:** 10.3390/e25020311

**Published:** 2023-02-08

**Authors:** Ignacio Reyes-Ayala, Marcos Miotti, Michal Hemmerling, Romain Dubessy, Hélène Perrin, Victor Romero-Rochin, Vanderlei Salvador Bagnato

**Affiliations:** 1Instituto de Fisica de São Carlos, Universidade de São Paulo, CP 369, São Carlos 13560-970, São Paulo, Brazil; 2Laboratoire de Physique des Lasers, Université Sorbonne Paris Nord, CNRS UMR 7538, F-93430 Villetaneuse, France; 3Instituto de Física, Universidad Nacional Autónoma de México, Apartado Postal 20-364, Ciudad de México 01000, Mexico; 4Biomedical Engineering Department, Texas A&M University, College Station, TX 77843, USA

**Keywords:** carnot cycles, BEC, critical phenomena, thermodynamics, quantum machines

## Abstract

Carnot cycles of samples of harmonically confined ultracold ${}^{87}$Rb fluids, near and across Bose–Einstein condensation (BEC), are analyzed. This is achieved through the experimental determination of the corresponding equation of state in terms of the appropriate global thermodynamics for non-uniform confined fluids. We focus our attention on the efficiency of the Carnot engine when the cycle occurs for temperatures either above or below the critical temperature and when BEC is crossed during the cycle. The measurement of the cycle efficiency reveals a perfect agreement with the theoretical prediction $\left(1-T_{L}/T_{H}\right)$, with $T_{H}$ and $T_{L}$ serving as the temperatures of the hot and cold heat exchange reservoirs. Other cycles are also considered for comparison.

## 1. Introduction

Thermodynamic transformations are among the most fascinating phenomena between the micro- and macro-world. We can think of many ways in which quantum effects can enter the work of thermodynamic engines. There are many considerations regarding quantum machines, either thermal [1] or not thermal [2]. When quantum fluids are involved, this is still new ground, which could be explored from many perspectives. The Carnot cycle is one of the keystones of thermodynamics, as it allows for the determination of temperature in absolute terms. This is provided by the Clausius inequality [3] which, for reversible processes, states that the ratio of absorbed to released heat, exchanged through the isothermal processes, can be identified using the ratio of the corresponding heat bath’s absolute temperatures, independently of the material cycle operating properties. At the same time, this result shows that the efficiency of the cycle is [4].
(1)$$\eta =1-\frac{T_{L}}{T_{H}}\;,$$
with $T_{H}$ and $T_{L}$ the temperatures of the hot and cold heat-exchange reservoirs. Since this universal result is independent of the medium carrying out the cycle, any correct thermodynamic description of any physical system realizing this cycle must obey it. There is no reason, in principle, to imagine that this consideration should not hold for a gas in the quantum regime [5]. Even in situations in which generic quantum features such as coherence and entanglement are present, the limit for efficiency still holds [6]. In light of these general statements, we present a thermodynamic analysis of an ultracold gas of ${}^{87}$Rb atoms confined in a harmonic trap, during which Carnot cycles are performed under different experimental conditions. Figure 1 shows the aim of our study:

For global-pressure Π vs. global-volume V, the appropriate thermodynamic variables for a harmonically confined gas [7,8], as explained below, we show three cycles: one within the normal phase $T>T_{c}$, one fully within the condensate phase $T<T_{c}$, and a third one, in which the quantum transition takes place within the cycle itself; namely, where the isothermal curves cross the Bose–Einstein condensation transition line. The latter, to the best of our knowledge, is the first experimental study of a cycle performed by a superfluid across its second-order phase transition (for theoretical studies, see [9,10]). As detailed below, we can directly calculate the efficiency of any cycle and compare this with its theoretical prediction, finding satisfying agreements. The type of analyses presented here are usually very difficult to perform for any fluid, if not impossible, due to the lack of accurate equations of state for the physical system at hand. In our case, we could achieve this goal because we had already developed a thermodynamic framework, Global Variables Thermodynamics [7,8], to properly describe the thermodynamic variables of an inhomogeneous trapped gas. In addition, after a very thorough experimental study on clouds of ${}^{87}$Rb, we obtained what we name a technical equation of state, from which the different processes in a cycle can be accurately obtained.

Thermodynamic cycles with single particles occupying discrete states of a potential [11,12,13] or many particles comprising a many-body quantum system [14] are a new concept within the topic of thermodynamic engines. The quantum effects in such systems may be used to compose relevant concepts in the new area of quantum technological devices [15]. In the same sense, the present work fits within the generic classification of “quantum engines” by using a quantum object as the main part of the system, allowing for quantum properties to lead to the occurrence of quantum phase transitions during the cycle’s performance. This is certainly a realistic scenario in the context of quantum engines. There are excellent reviews discussing many theoretical aspects of quantum thermodynamic devices [16].

Fluids confined in non-uniform traps, such as those in harmonic confinement, are also inhomogeneous. Hence, the hydrostatic pressure becomes locally spatial-dependent and the volume is not defined. As such, pressure and volume are no longer thermodynamic variables. Based on this observation, in recent years, we developed the correct thermodynamic description, called Global Thermodynamics, by identifying the proper equilibrium mechanical variables “pressure” and “volume” [8], which we called global pressure (Π) and global volume (V). This identification follows from the formal expression of statistical physics of free energy and entropy in the thermodynamic limit. For instance, as described in detail in Ref. [8], a gas of *N* neutral atoms can be accurately described by a Hamiltonian with pairwise interactions confined by an external field, as follows:(2)$$H=\sum_{i=1}^{N}\frac{p_{i}^{2}}{2m}+\sum_{i<j}u(|r_{i}-r_{j}\left|\right)+\sum_{j=1}^{N}V_{\mathrm{ext}}\left(r_{j}\right),$$
with $u(|r_{i}-r_{j}|)$ is a pairwise, short-range, interatomic potential and the external field, in our case, is a harmonic trap of optical and/or magnetic origin [17,18],
(3)$$V_{\mathrm{ext}}\left(r\right)=\frac{1}{2}m\left(\omega_{x}^{2}x^{2}+\omega_{y}^{2}y^{2}+\omega_{z}^{2}z^{2}\right)\;,$$
with r=(x,y,z) and $\omega_{i}$ as the frequencies of the confining trap potential. It can be rigorously shown that, by appealing to the definition of the free energy of a gas at temperature *T*,
(4)$$F=-kT\ln\mathrm{Tr}\;e^{-\beta H}\;,$$
in the appropriate thermodynamic limit, N→∞, V→∞ with N/V= constant, the Helmholtz free energy is found to be an extensive function F=F(N,V,T), where the global extensive volume is given by $V=\frac{1}{\omega_{x}\omega_{y}\omega_{z}}$. Accordingly, the conjugate intensive global pressure is given by:(5)$$\begin{array}{ccc} \Pi & = & -{\left(\frac{\partial F}{\partial V}\right)}_{N,T}\\ & = & \frac{1}{3V}\int \;r\cdot \nabla V_{ext}\left(r\right)\;\rho \left(r\right)d^{3}r. \end{array}$$
The second line is a generalization of the hydrostatic pressure in terms of the virial of the external force and, essentially, the equation for force equilibrium within the fluid. However, more importantly for application purposes, the second line also provides a procedure that can be used to determine the global pressure Π as an integration of the inhomogeneous density profile ρ(r), which is weighted by the confining external potential. It is also important to recall that all thermodynamic formalisms follow from, for instance, the usual expression for the free energy, F=U−TS=−ΠV+μN and dF=−SdT−ΠdV+μdN. In this way, the work involved in any reversible process is given by:(6)$$W=-\int_{A}^{B}\Pi \;dV\;,$$
where the integral occurs along the given process from state *A* to state *B*. It is worth noting that, if the external potential $V_{ext}\left(r\right)$ is not harmonic, although it is a confining potential, one can always find the appropriate generalized or global volume V to such an external potential ([8]) and, once it is identified, the definition of the corresponding pressure Π is given by (Equation 5).

## 2. Cycles

Using these global variables, we can set up any cycle in the usual Π-V diagram, at constant *N*. The Carnot cycle is defined by four reversible processes [19]. Starting in an initial state $\left(\Pi_{1},V_{1},T_{H}\right)$, an isothermal expansion takes place to volume $V_{2}>V_{1}$, while heat is absorbed from the reservoir at $T_{H}$. Then, a further adiabatic expansion to $V_{3}$ cools the gas to $T_{L}$. This is followed by an isothermal compression to $V_{4}$, releasing heat to the reservoir at $T_{L}<T_{H}$. Finally, the system returns to its initial state with adiabatic compression. The efficiency of a thermodynamic cycle is the ratio of the work *W* carried out by the system to the heat absorbed $Q_{H}$. According to the Second Law of Thermodynamics, it is true that the efficiency depends only on the isotherm temperatures in the cycle [20], as follows:(7)$$\eta =\frac{W}{Q_{H}}=1-\frac{T_{L}}{T_{H}}\;.$$

The main goal of our experiment is to predict the work that the system can complete and the corresponding absorbed heat in each ideal cycle. These can be used to find the efficiency and, a posteriori, verify that the efficiency equals the expected expression. The work performed during the whole cycle can be calculated if one uses the parametrization of the isothermal and adiabatic curves in the Π-V diagram, at constant *N*. The former are directly obtained from the equation of state $\Pi =\Pi \left(N,V,T\right)$, while the latter can be derived from the relationship of the entropy *S* as a function of $\left(N,V,T\right)$,
(8)$$TdS=C_{V}dT+T{\left(\frac{\partial \Pi }{\partial T}\right)}_{N,V}dV$$
where $C_{V}$ is the heat capacity at constant *N* and V. In a previous work by our group [21], we showed that the heat capacity can be accurately approximated by $C_{V}=3V{\left(\frac{\partial \Pi }{\partial T}\right)}_{N,V}$. The validity of this expression rests on the fact that density profiles below BEC are basically fitted by a bimodal distribution. In this way, the internal energy can be separated into two parts: one arising from the thermal cloud and the other from the condensate fraction. In turn, the global pressure of the condensate is much smaller than the pressure of the thermal cloud; following condensation, the expression for $C_{V}$ holds exactly. Hence, the adiabatic curves can be found by setting dS=0 in Equation (Equation 8), yielding $VT^{3}=\mathrm{constant}$; therefore, with the use of the equation of state, we can find Π as a function of V along the adiabatic curves. To calculate the heat absorbed along the isotherm at $T_{H}$, we integrate Equation (Equation 8) for constant *T*,
(9)$$Q_{H}=\int_{1}^{2}TdS=\int_{1}^{2}{\left(\frac{\partial \Pi }{\partial T}\right)}_{N,V}dV\;.$$

In a typical experimental run, about $10^{5}$ rubidium atoms (${}^{87}$Rb) are confined in a magnetic trap, where the sample is cooled to the order of a few microkelvins through radiative cooling methods [22]. Then, with radiofrequency methods, evaporation cools the atomic cloud down cools down to around 100 nK, with $∼10^{4}$ atoms in the Bose–Einstein condensate. Variations in controllable elements, such as heat and range of evaporation frequencies, magnetic fields and number of atoms, allow for the global thermodynamic variables *N*, Π and V to be mapped at different values. The global volume $V=1/\omega_{x}\omega_{y}\omega_{z}$ is known independenly from accurate measurements of the trap frequencies, using center-of-mass oscillations. With the knowledge of these frequencies, the in situ spatial density distribution can be reconstructed from the absorption time-of-flight images [23] of the trapped atomic cloud. Then, from this image, we can obtain the temperature *T*, number of particles *N* and global pressure Π; the temperature was found by fitting the density profile tail to the expected Gaussian equilibrium profile of a diluted gas; the number of particles *N* were found using the absorption optical density of the image; the global pressure Π was calculated using the theoretical expression (Equation 5). We recall that the simplest criterion that was used to identify if the gas suffered from Bose-Einstein condensation, namely, if $T<T_{c}$, was the appearance of a bimodal density distribution with a Thomas–Fermi peak in the center and a thermal Gaussian tail; in the normal gas phase, $T>T_{c}$, the density profile is a broad Gaussian thermal function. One can further verify that this is correct by rotating the sample and observing the appearance of superfluid quantized vortices in the condensed phase [17]. More details on the experimental set-up and procedures can be found in previous works [18,21]. Then, to construct the equation of state $\Pi =\Pi \left(N,V,T\right)$, as described below, we first generated a large amount of dataset with nearly 500 density profiles ρ(r). Each of these yielded the thermodynamic variables (N,V,T,Π) for each experimental thermodynamic state.

The equation of state Π=Π(N,V,T) was found by a fitting procedure. We propose a so-called technical equation for the global pressure as a function of temperature, with five technical coefficients ($a_{i}$) that depend on the number of particles *N* and the global volume V,
(10)$$\Pi \left(N,V,T\right)=\left\{\begin{array}{ccc} a_{0}\left(N,V\right)T^{4+a_{1}\left(N,V\right)}+a_{2}\left(N,V\right) & \; & T\leq T_{c}\\ a_{3}\left(N,V\right)T+a_{4}\left(N,V\right) & \; & T\geq T_{c} \end{array}\right.$$
Above the critical temperature, we fit an ideal gas type of equation, while below condensation, we fit the equation of a Bose gas using a harmonic potential corrected by a coefficient to consider the first-order atomic interactions, since the gas is in a diluted regime. An equation of this form is proposed, since a previous finding indicated that the global pressure depended on temperature, with a power law close to 4 below the critical temperature [24]. Above this, we can recover the ideal Bose gas equation. It is interesting to note that the pressure does not reach zero when the temperature is zero; that is, there is a remaining zero-point pressure [25]. To find the technical coefficients $a_{i}$, the experimental points of a typical run, such as those in Figure 2, are fitted by Equation (Equation 10) where one of the thermodynamic variables must be fixed: either the number of particles *N* or global volume V. We chose to fix the volume, as this is a controllable variable in our experiment. Hence, we ended with a collection of values of (N,T,Π) for each volume V, from which we can study any desired cycle. Although the pressure should be intensive, such as $\Pi =\Pi \left(N/V,T\right)$, we did not assume this at the outset; however, the results show that this was indeed the case—see Ref. [26] for further details.

Using the experimental technical equation of state, we constructed Carnot cycles in several different conditions, such as those shown in Figure 1. As described, the cycles can be constructed considering different adiabatic and isothermal curves. Using those, we can calculate the work performed in the cycle and the absorbed heat and, in turn, we can then calculate the experimental efficiency of the cycle. Figure 3 shows a plot representing the measured efficiency $\eta =\frac{W}{Q_{H}}$ plotted against the theoretical Carnot efficiency $\eta =1-\frac{T_{L}}{T_{H}}$ when the critical temperature is not crossed. Several different temperatures in the cold and hot reservoirs, $T_{L}$ and $T_{H}$, were considered. As we can see, the points that represent each cycle fall very close to the identity curve, proving that the experimental efficiency equals the theoretical thermodynamic efficiency of a Carnot cycle. Note that, for temperatures above the critical line of temperatures $T_{c}$, the density profile occurs in the thermal phase; below this, there is a combined condensed phase and thermal phase.

In Figure 4, we show the cycle efficiency $\eta =\frac{W}{Q_{H}}$ versus the Carnot efficiency $\eta =1-\frac{T_{L}}{T_{H}}$ for a series of mixed cycles that do cross the transition line. This is quite an interesting case, as it proves the prediction of the Second Law: no matter the engine material or the equilibrium states, the efficiency depends only on the temperatures of the reservoirs that absorb and release heat during the isothermal processes. This occurs regardless of whether there is second-order phase transition between two macroscopic quantum fluids, as expected, as other thermodynamic properties of the working medium show that a transition occurs during the cycle. This is illustrated in Figure 5, where we show the behavior of the global isothermal compressibility $K_{T}$ during the cycle. The compressibility becomes discontinuous when crossing the critical line.

For completion, in Figure 6 and Figure 7 we show a comparison of Carnot cycles versus Otto and Ideal cycles and their respective efficiencies. The first is composed of two adiabatic processes, connected by two isochoric processes. This cycle is of particular interest in the study of quantum thermodynamic engines in recent experimental systems [27,28], while the second is composed of two isobaric processes connected by two isochores. The three cycles operate between the same extreme temperatures, $T_{H}$ and $T_{L}$, as indicated in Figure 6. Evidently, as shown in Figure 7, Carnot cycles yield the highest efficiency for equivalent conditions.

The successful combination of global thermodynamics and one of the most fundamental predictions of the Second Law, namely, the efficiency of a Carnot cycle, can be used in the correct identification of the mechanical thermodynamic properties, global pressure and volume of a harmonically confined ultracold gas. In addition to verifying that the work performed by a confined gas consists of changing the trap frequencies, keeping other variables, such as entropy or temperature, constant, global thermodynamics provides a proper expression for this basic and fundamental quantity. In this way, the First Law is also clearly stated since, independently of whether a system is confined, energy and heat can always be unambiguously determined, allowing for us to generally write (at constant *N*), dE=TdS−ΠdV.

To conclude, we analyzed thermal engine cycles that operate above or below Bose–Einstein condensation and cross the superfluid transition line. As previously stated, this type of analysis is generally very difficult to perform, since one needs accurate equations of state. Our study, showing the strength of the technical equation of state for the thermodynamic description of confined ultracold gases, motivates further theoretical quantum many-body research to justify and improve this description, and opens the door for research on the potential implementations and applications of actual thermal engines at a macroscopic quantum level. Finally, verification of the use of quantum material in the thermal engine, exploiting possible quantum effects, shows the maximum possible efficiency of the thermal engine. This verifies that, within the described context, the Carnot efficiency still holds. The inclusion of quantum phases, such as Bose–Einstein condensates, in the thermal engines introduces a new type of calorimetry in the thermodynamics of closed-cycle transformations.

## Figures and Tables

**Figure 1 entropy-25-00311-f001:**
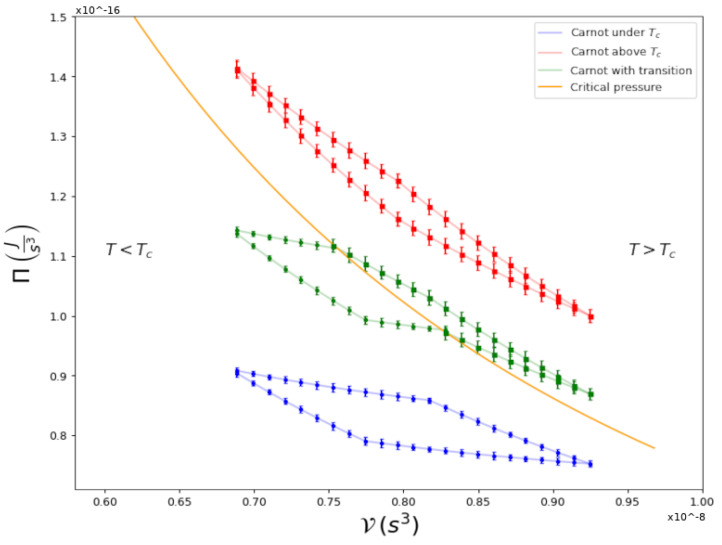
Carnot Cycles in a Π-V diagram for a ${}^{87}$Rb gas that is harmonically trapped.

**Figure 2 entropy-25-00311-f002:**
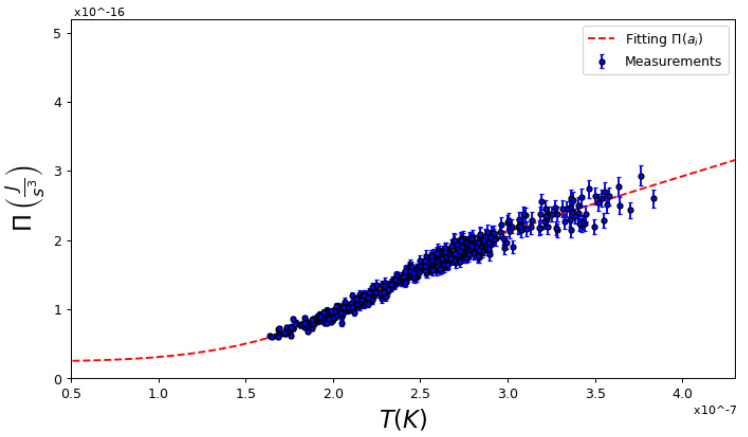
Typical dependence of values for global pressure and fitting using Equation (Equation 10). While above the critical temperature the system is a linear curve, below this, it is close to a fourth-order dependence with *T*. ($N=4.2\times 10^{5}$, $V=7.8\times 10^{-9}\;s^{3}$, $T_{c}=2.5\times 10^{-7}\;K$).

**Figure 3 entropy-25-00311-f003:**
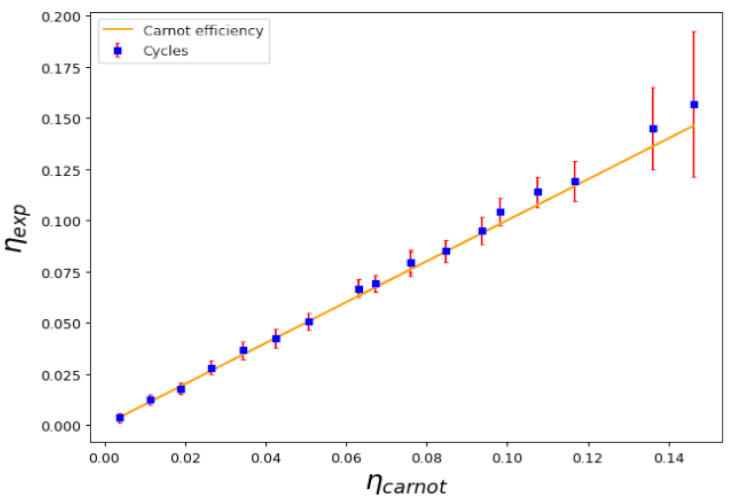
Calculated efficiency $\eta =\frac{W}{Q_{H}}$ versus Carnot efficiency $\eta =1-\frac{T_{L}}{T_{H}}$ for a series of cycles above and below the critical temperature; namely, those with no transition during the cycle. Despite having a condensed phase, the efficiency of the cycle maintains the expected efficiency for a Carnot cycle.

**Figure 4 entropy-25-00311-f004:**
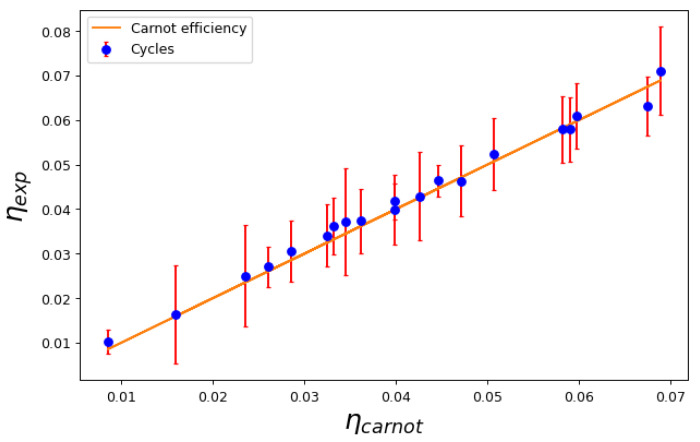
Calculated efficiency $\eta =\frac{W}{Q_{H}}$ versus Carnot efficiency $\eta =1-\frac{T_{L}}{T_{H}}$ for a series of cycles that cross the transition line.

**Figure 5 entropy-25-00311-f005:**
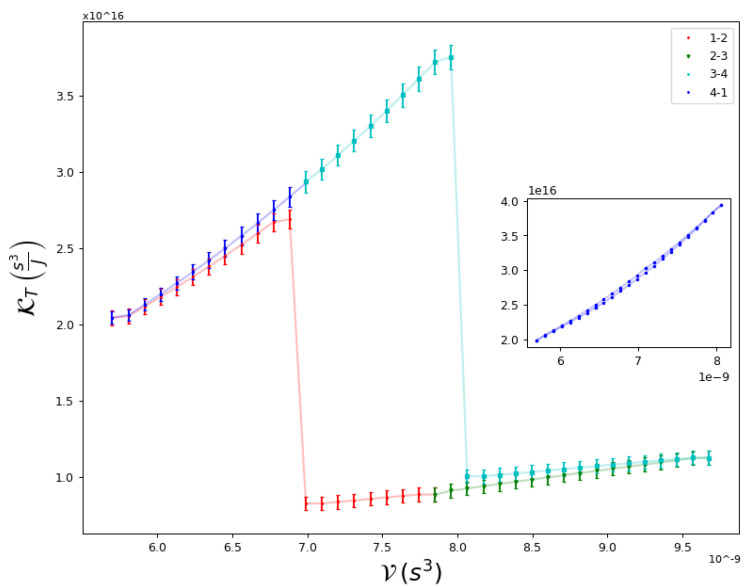
Global isothermal compressibility $K_{T}=-\frac{1}{V}{\left(\frac{\partial V}{\partial \Pi }\right)}_{N,T}$ as a function of V along a Carnot cycle, whose isothermal processes cross the transition line. These crosses are observed as discontinuities in the compressibility. In the inset, the compressibility is shown for a cycle that does not cross the transition.

**Figure 6 entropy-25-00311-f006:**
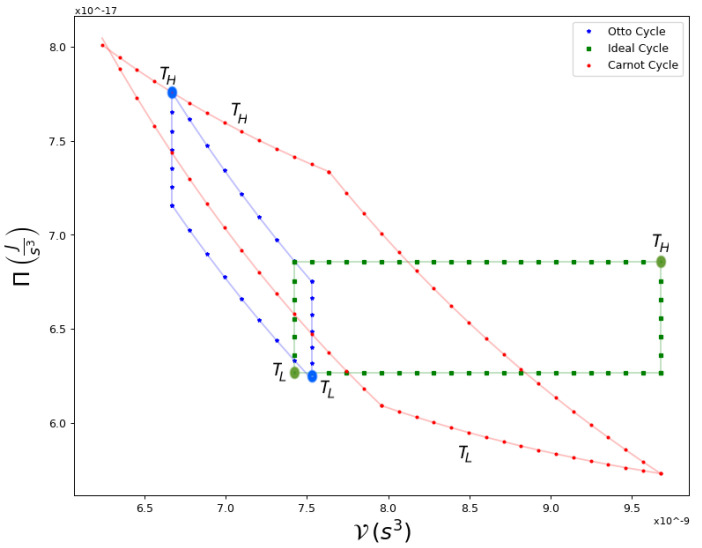
An example of Ideal, Otto and Carnot Cycles in a Π-V diagram, within the same extreme temperatures, marked with large dots, $T_{H}$ and $T_{L}$. These are the same hot and cold temperatures that occur for the Carnot cycle. The Ideal cycle is composed of two isobaric curves and two isochores, while the Otto cycle has two adiabatic processes and two isochoric ones.

**Figure 7 entropy-25-00311-f007:**
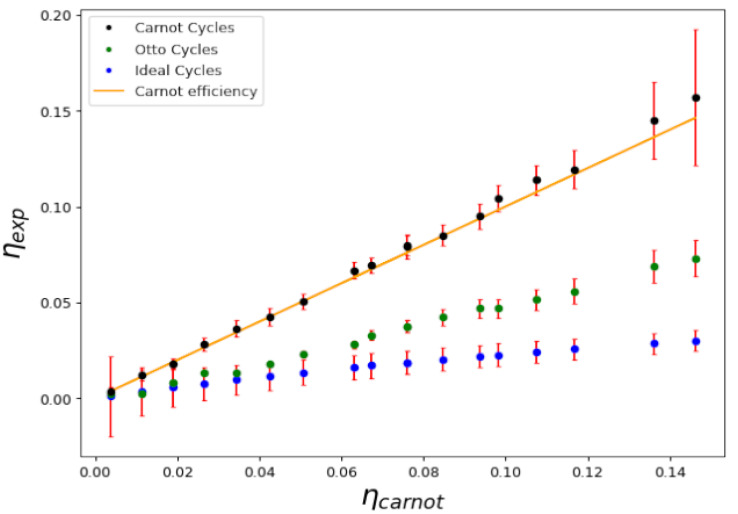
Ratio of the calculated efficiency to the Carnot one for a series of ideal, Otto and Carnot cycles, as a function of experimental efficiency. The ideal cycle efficiency is approximately 0.2; the Carnot efficiency and the Otto cycle efficiency are 0.3.

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
