# Peer review of "Carnot Cycles in a Harmonically Confined Ultracold Gas across Bose–Einstein Condensation"

_entropy, 2023, doi:10.3390/e25020311_

Round 1

Reviewer 1 Report

see pdf attached

Author Response

We appreciate your comments and suggestions for improvement our manuscript. In an attached file, we include the answer.

Reviewer 2 Report

The paper is interesting and can be eventually published.

The style of the text is not helping to the good understanding by the reader.

There are too many references to previous works and the reader interested, after reading the abstract, in the results of the current work may not have enough time to follow the numerous references. I would recommend authors to more carefully explain the experiment, how the phase transition is observed, why the volume and pressure are defined in a certain way etc. by the authorsThe "global thermodynamics" has to be defined clearly instead of a sentence that it is in confrontation with the second law. I will not give more examples as the whole text has to be carefully reconsidered.

Special attention should be given to the language that is unsatisfactory in the current version. Already in the Introduction we see "mean medium" (what is it?), then the T_{h} appears without any explanation, the Clausius name is written in a wrong way, "for comparison"  twice in the same sentence, "final remarks ... is given", three times "efficiency" in two lines of the same sentence, etc.  Many sentences are to be first read by the authors as if this would be done by a new reader and then it is easy to see that the whole style has to be rewritten. This should be done with the entire text, many sentences are written not carefully, both from the viewpoint of English, but mainly to take care of a reader (otherwise the reader can be repelled by the inaccuracies of the text). There are many misprints ("sing" instead of "sign",...) and sentences with unclear meaning. Special care should be applied to equations. Look, for example, at eq. (15), it contains two terms of identical structure but after some time you understand that there is just not a good writing and the notation $T$ should be separated from the preceding symbols by a comma or something like that.

The paper can be accepted only after this "dirty work" of rewriting would be performed having in mind not just a report about some performed activity, but the interests of the future reader.

Author Response

(The authors gave the same response as above.)

Round 2

Reviewer 1 Report

Although it wasn't exactly my objection, I noticed that the authors made considerable improvements in the presentation of the work with regard to the formal aspect, making the manuscript self-sufficient. Besides, in the new version the authors now explicitly say, when referring to Fig.7, that "Evidently, as shown in Fig. 7, Carnot cycles yield the highest efficiency for equivalent conditions."  However, they did not answer my question. As I had pointed out, if all cycles are done reversibly and under two reservoirs with the same two temperatures, all cycles must have the same efficiency. In the authors' response, however, they seem to say that the Carnot cycle may have greater efficiency than other reversible cycles, which is not true. Indeed, as we know from Carnot's theorem, this contradicts Clausius' statement of the second law. Although I am willing to recommend the manuscript, in my opinion this point is very important and should not be overlooked.

Author Response

Dear Referee, find attached our answer.

Reviewer 2 Report

The paper in the current version can be accepted.

One question remains not explained: what should be taken as the effective volume if the trap is not harmonic? There should be a general recipe independent of the exact trap potential. Authors should at least say something on this problem.

Author Response

We thank Referee 2 for its reading of the paper and for considering it publishable.

As far as his/her question regarding the generalization or “recipe” for finding an expression for the global volume for an arbitrary confining potential, indeed in Ref. 8 this issue is discussed and it is shown how such a global or generalized volume can be found. We have added a sentence in the paper, after equation (6), to point out this important result.

Round 3

Reviewer 1 Report

The authors have responded satisfactorily to the points I have raised. In my opinion the manuscript is now ready for publication.